# TREEFORMER: DENSE GRADIENT TREES FOR EFFICIENT ATTENTION COMPUTATION

**Lovish Madaan, Srinadh Bhojanapalli, Himanshu Jain & Prateek Jain**
Google Research
{lovishm,bsrinadh,himj,prajain}@google.com

## ABSTRACT

Standard inference and training with transformer based architectures scale quadratically with input sequence length. This is prohibitively large for a variety of applications especially in web-page translation, query-answering etc. Consequently, several approaches have been developed recently to speedup attention computation by enforcing different attention structures such as sparsity (Zaheer et al., 2020), low-rank (Wang et al., 2020), approximating attention using kernels (Choromanski et al., 2021). In this work, we view attention computation as that of nearest neighbor retrieval, and use decision tree based hierarchical navigation to reduce the retrieval cost per query token from linear in sequence length to nearly logarithmic. Based on such hierarchical navigation, we design *Treeformer* which can use one of two efficient attention layers – TF-ATTENTION and TC-ATTENTION. TF-ATTENTION computes the attention in a fine-grained style, while TC-ATTENTION is a coarse attention layer which also ensures that the gradients are "dense". To optimize such challenging discrete layers, we propose a two-level bootstrapped training method. Using extensive experiments on standard NLP benchmarks, especially for long-sequences, we demonstrate that our TREEFORMER architecture can be almost as accurate as baseline Transformer while using 30x lesser FLOPs in the attention layer. Compared to Linformer, the accuracy can be as much as 12% higher while using similar FLOPs in the attention layer.

## 1 INTRODUCTION

Self attention layer is the key component of Transformers (Vaswani et al., 2017), enabling them to achieve state of the art performance across tasks in Natural Language Processing (Devlin et al., 2019; Radford et al., 2019; Raffel et al., 2019) and Computer Vision (Dosovitskiy et al., 2021). Attention computation scales quadratically ($n^2$) with the input sequence length ($n$), making it a key bottleneck in scaling Transformers to long inputs. This has resulted in a lot of alternate proposals to efficiently compute attention using different approximations. However, as shown in Tay et al. (2021), methods that offer dramatic speedups (Wang et al., 2020; Choromanski et al., 2021) suffer in accuracy, and methods that match accuracy of Transformer (Zaheer et al., 2020) don't offer significant speedups.

In this paper we develop a novel efficient attention mechanism using *input dependent sparsity structure* in attention computation, motivated by the analysis showing the sparse nature of attention (Shi et al., 2021; Gupta et al., 2021). We propose to use *decision trees* to efficiently compute attention by only retrieving the top *nearest neighboring keys* for a given query. Decision trees are low cost, hierarchical non-linear models that have been popular in both supervised (Chen & Guestrin, 2016; Quinlan, 2014) and unsupervised (Duda et al., 1973) learning. Given a vector, each node of the decision tree decides whether to navigate to the left or the right child depending on the sign of a simple linear projection. We learn decision trees in each attention layer of a Transformer model so that a query pays attention only to keys that map to the same leaf nodes[1] of the decision trees. We refer to this as TF-ATTENTION. Such sparse decision trees that retrieve only a single leaf node are typically hard to train due to their discrete nature. Further, queries that pay strong attention to multiple keys result in bunching many keys together in a single leaf node creating imbalance. The key computational advantage of decision trees is realized only when they are balanced.

---

[1]Leaf nodes are the bottom nodes of a decision tree with 0 children.

To address these issues we propose another attention mechanism TC-ATTENTION, that in addition to keys in the leaf node, pays attention to keys at all the intermediate nodes traversed by the query. We further combine the retrieved value vectors with query independent, but trainable weights. This allows us to learn a decision tree that clusters keys in a hierarchical and balanced manner. In both these methods we train the decision tree in an end-to-end manner with the rest of the components of the Transformer. Since the decision trees themselves are inexpensive to evaluate, this allows us to reduce attention computation cost dramatically, requiring typically only a few dot products per query.

In our goal to train these decision trees in an end to end manner with the rest of the Transformer components we faced some challenges. Naively using the decision trees to restrict attention computation results in poor optimization with training getting stuck at high loss. To solve this we introduce a novel bootstrapping method that allows us to gradually restrict attention computation in layers using decision trees. While this keeps training expensive initially, it still dramatically reduces inference costs. We further use *Dense Gradient Trees* (Karthikeyan et al., 2021), allowing us to use simple gradient based optimization.

We evaluate the proposed architecture on popular NLP models such as BERT (Devlin et al., 2019), and on the Long Range Arena benchmark (LRA) (Tay et al., 2021), developed specifically to compare efficient attention methods on long sequence length tasks. We show that TREEFORMER is able to match/improve the performance over other popular models while achieving lower computational cost, for e.g. - TREEFORMER has 8-9x lesser FLOPs than models like BigBird (Zaheer et al., 2020) and Performer (Choromanski et al., 2021).

Our contributions in this paper are — **(i)** We pose attention computation as a nearest neighbor retrieval problem, and propose a novel architecture TREEFORMER − using dense gradient trees for retrieving top keys to pay attention to for a given query. We develop two novel attention mechanisms using decision trees - that, for a given query, pay attention to only keys in the matching leaf node (TF-ATTENTION) or to all the keys along the traversed path (TC-ATTENTION). **(ii)** We develop a novel bootstrapping method to gradually sparsify attention computation thereby showing that decision trees can be trained as a part of neural networks using back propagation. **(iii)** We experimentally show the effectiveness of the proposed architecture using BERT models and on LRA (Tay et al., 2021). In particular, both TF-ATTENTION and TC-ATTENTION have matching accuracy to baseline Transformer architecture, but with up to $30\times$ cheaper (in FLOPs) attention layer compared to standard Transformers and up to $9\times$ cheaper attention layer compared to SOTA BigBird (Zaheer et al., 2020).

## 2 PRELIMINARIES: ATTENTION AND DECISION TREES

### 2.1 ATTENTION

Let $\boldsymbol{Q}, \boldsymbol{K}, \boldsymbol{V} \in \mathbb{R}^{n \times d}$ be the input query, key, and value matrices, where $n$ is the sequence length and $d$ is the embedding dimension, and $W^Q, W^K, W^V \in \mathbb{R}^{d \times d}$ are their respective projection matrices. The standard attention in each Transformer layer can be defined as

$$\text{Attention}(\boldsymbol{Q}, \boldsymbol{K}, \boldsymbol{V}) = \text{softmax}\underbrace{\left[\frac{\boldsymbol{Q}W^Q(\boldsymbol{K}W^K)^T}{\sqrt{d}}\right]}_{\boldsymbol{A}} \cdot \boldsymbol{V}W^V \tag{1}$$

Note that for the self attention layer, all the three matrices $\boldsymbol{Q}, \boldsymbol{K}, \boldsymbol{V}$ are same as the input to the layer. Computing both attention $\boldsymbol{A}$ and its product with the projected value matrix $\boldsymbol{V}W^V$ is an $O(n^2 d + d^2 n)$ operation. In particular, assuming $cpqr$ cost for multiplying two matrices of size $p \times q$ and $q \times r$, the main cost of this layer is: $c(2n^2 d + 2d^2 n)$.

This computation becomes a major bottleneck in applications which have large sequence lengths $n$. To mitigate this, several efficient attention mechanisms have been proposed (Child et al., 2019; Wang et al., 2020; Choromanski et al., 2021; Kitaev et al., 2020; Sun et al., 2021; Wang et al., 2022). Child et al. (2019) proposed sparse attention ($\boldsymbol{A}$) with different sparsity patterns that reduce attention cost from $O(n^2)$ to $O(n^{1.5})$. Later Zaheer et al. (2020); Yun et al. (2020) proved universal approximation power of Transformers with sparse attention, improving the cost to $O(ns)$, with constant sparsity $s$ per query. Alternately Wang et al. (2020) considered a low rank approximation to attention, reducing computation cost to $O(nk)$ for rank $k$ approximation. Choromanski et al. (2021) considered a kernel

based approach to approximate attention computation reducing the cost to $O(nr)$, where $r$ is the dimension of the random features used to compute the kernel. We refer readers to Tay et al. (2020) for a detailed survey of efficient attention methods. In this paper, we propose a decision tree based mechanism to speed up the attention computation to $O(nh)$, where $h$ is the height of the decision tree. We show that with TREEFORMER even a $h$ as small as 8 is sufficient for most applications, leading to the least amount of FLOPs required, while maintaining the performance.

Table 1: Comparison of attention computation cost (eq.equation 1) for different models.

| Model | Transformer | Linformer | BigBird | Performer | TREEFORMER TF-A |
|---|---|---|---|---|---|
| Attention cost | $O(n^2d + n^2)$ | $O(knd + kn)$ | $O(snd + sn)$ | $O(rnd + n(r + d))$ | $O(hnd + (2^{h+1} - 1)d)$ |

## 2.2 DECISION TREES

Decision trees are a classical technique that enables partitioning of the input space in several oblique blocks that can be navigated efficiently using the hierarchical structure. Decision trees and similar hierarchical structures are highly popular in supervised learning (XgBoost (Chen & Guestrin, 2016), C4.5 (Quinlan, 2014)), unsupervised learning (hierarchical clustering (Duda et al., 1973)), and retrieval (KD tree (Bentley, 1975)). Generally, hierarchical retrieval techniques are nominally data-dependent, e.g., PCA-tree (Sproull, 1991) and Cover Trees (Beygelzimer et al., 2006). Our TC-ATTENTION and TF-ATTENTION methods combine hierarchical retrieval methods with *supervised* tree learning techniques to enable learning of the hierarchical attention layer in end-to-end differentiable manner. See Figure 1 for overall representation of the hierarchical navigation based attention layer. Each internal node of the tree has a classifier that determines which children a given input should navigate to. While the classifier makes hard decision, we use technique from straight-through estimators (Bengio et al., 2013), and in particular, *Dense Gradient Trees* from Karthikeyan et al. (2021) to train the tree parameters using gradient style methods.

**Construction** Formally, let $\mathcal{T}(\theta)$ represent a binary tree with height $\boldsymbol{h}$ and $2^{\boldsymbol{h}}$ nodes. Let $\theta_{\{l,j\}}$ denote the tree parameters at $\{l, j\}$-th node where $l \in \{0, 1, \cdots, \boldsymbol{h}\}$ is the node level and $j \in \{0, 1, \cdots, 2^l - 1\}$ is the node index at level $l$. Let decision at each internal node to send the point $\boldsymbol{q}$ to left or the right child is based on $\text{SIGN}(f(\theta_{\{l,j\}}; \boldsymbol{q}))$, where,

$$f_{\{l,j\}}(\boldsymbol{q}) = f(\theta_{\{l,j\}}; \boldsymbol{q}) = \langle \boldsymbol{w}_{\{l,j\}}, \boldsymbol{q} \rangle + b_{\{l,j\}}, \tag{2}$$

and $\theta_{\{l,j\}} = \{\boldsymbol{w}_{\{l,j\}}; b_{\{l;j\}}\}$ consists of a oblique hyperplane and a bias. Let $P_{\mathcal{T}}(l, \boldsymbol{q})$ denote the node-index of $\mathcal{T}$ that was traversed by $\boldsymbol{q}$ at level $l$. That is, $P_{\mathcal{T}}(h, \boldsymbol{q})$ denote the leaf node to which $\boldsymbol{q}$ is mapped. Also, let $S_{l,j}(\boldsymbol{K})$ denote the indices of vectors from the key matrix $\boldsymbol{K}$ that traverses node $\{l, j\}$. That is, $i \in S_{l,j}(\boldsymbol{K})$ iff $j = P_{\mathcal{T}}(l, \boldsymbol{k}_i)$.

## 3 TREEFORMER ATTENTION VARIANTS

In this section, we describe the decision tree attention mechanism which replaces the standard dot product attention (Vaswani et al., 2017) in our TREEFORMER architecture. Recall that for each query $\boldsymbol{q}_i$, the attention is given by:

$$\boldsymbol{A}_i = \text{softmax}(\frac{1}{\sqrt{d}}\boldsymbol{q}_i^T W^Q (W^K)^T \boldsymbol{K}^T) \boldsymbol{V} W^V.$$

That is, attention computation for $\boldsymbol{q}_i$ is equivalent to applying a *linear scan* over all keys, and computing value as the weighted average of value of most *similar* keys to $\boldsymbol{q}_i$. Recent works have shown that we do not need to attend to all tokens in a given input sequence (Shi et al., 2021), and demonstrate that *top-k attention* is a good approximation for standard attention (Gupta et al., 2021).

We propose to leverage this observation by using a decision tree based hierarchical navigation instead of linear scan to find the top-most *similar* keys. Naturally, accuracy of the method would depend on accuracy of retrieval of most similar keys by the hierarchical navigation method. So, we propose to learn the decision tree parameters so as to explicitly optimize the loss function. We develop two approaches to compute the value for a query $\boldsymbol{q}_i$ based on its traversal through the tree. In the following subsections we present our two different tree based attention layers.

### 3.1 TF-ATTENTION (*Tree Fine-grained Attention*)

In this variant, we compute the value for a query $\boldsymbol{q}$ as a weighted sum of value vectors of keys that lie in the same leaf node as $\boldsymbol{q}$. More formally,

$$\text{TF-ATTENTION}(\boldsymbol{q}, \boldsymbol{K}, \boldsymbol{V}; \mathcal{T}) = \text{softmax}\left[\frac{\boldsymbol{q}^T W^Q (\boldsymbol{K}_{\bar{S}} W^K)^T}{\sqrt{d}}\right] \cdot \boldsymbol{V}_{\bar{S}} W^V, \qquad (3)$$

where $\bar{S} = S_{h, P_{\mathcal{T}}(h, (W^Q)^T \boldsymbol{q})}(\boldsymbol{K} W^K)$ is the set of keys $\boldsymbol{K}$ that are mapped to same leaf node (i.e. $P_{\mathcal{T}}(h, (W^Q)^T \boldsymbol{q})$) as $(W^Q)^T \boldsymbol{q}$. Note that this is a sparse attention variant as we compute dot products only between those queries and keys belonging to the same leaf node, and the attention score between a query and keys lying in different leaf nodes is 0. Intuitively, this attention layer can be thought of as top-$k$ attention where the tree structure is responsible for computing the $k$ tokens which are the most important for a given query token.

**Computational Complexity** Assuming a uniform distribution of the input sequence tokens across the leaf nodes of the decision tree, the amortized computational cost for TF-ATTENTION is $O(\frac{n^2 d}{2^h}) + 2cd^2 n \equiv O(nkd) + 2cd^2 n$, where $k$ is generally a small factor assuming the height of the decision tree is of the order $O(log(\frac{n}{k}))$, and $c$ is as defined in Section 2.1.

If the decision tree parameters are such that all the queries and keys get clustered into a single leaf node of the tree, we get the original full attention in this case. Therefore the worst case time complexity of TF-ATTENTION is $O(n^2 d + d^2 n)$. Furthermore, this also illustrates that TF-ATTENTION can be as expressive as standard attention. We state this claim formally below.

**Lemma 1.** *There exists a tree $\mathcal{T}$ s.t.* TF-ATTENTION$(\boldsymbol{Q}, \boldsymbol{K}, \boldsymbol{V}; \mathcal{T}) = Attention(\boldsymbol{Q}, \boldsymbol{K}, \boldsymbol{V})$.

The lemma follows immediately from the above observation; for completeness, see Appendix for a proof of the above lemma.

$k$-**ary TF-ATTENTION** We extend the binary tree construction of the TF-A variant to construct a $k$-ary tree attention model for height $h$, where each node at level $l$ has a branching factor of $b_l$, $l \in \{0, 1, \cdots, h-1\}$. The total number of leaf nodes in this case are $\prod_{l=1}^{l=h-1} b_l$. This is a general version of the TF-ATTENTION variant, where we can scale the number of leaf nodes even for smaller heights, and thus reducing the amount of queries and keys in each leaf node. In all our experiments on $k$-ary TF-ATTENTION models, $h = 2$ and $b = [100, 10]$, resulting in 1000 leaf nodes.

Although the proposed TF-ATTENTION TREEFORMER variant works well in principle, there are a couple of shortcomings of the same — (i) In case the distribution of queries and keys is skewed in even one of the layers of the model, the computational cost approaches the worst case time complexity of the standard attention model. So we might need to add explicit loss terms to encourage more uniform representation across various leaf nodes/subtrees. (ii) It might be possible that for a given query, some key tokens important to this query lie in a different leaf node of the decision tree to minimize the overall loss. But due to the sparse nature of the model, the attention score between the query and these keys will be 0. Using multiple decision trees (forest structure) might help alleviate this but we leave it for future exploration.

We address the above problems using a novel variant that computes the attention values in a *coarse* manner and hence has denser gradients.

### 3.2 TC-ATTENTION (*Tree Coarse Attention*)

In this variant, rather than computing attention as a weighted sum of values of keys in the leaf node, we take a *coarse* unweighted sum of the value vectors. That is, the query is only used to navigate to a leaf node, but with-in a leaf node, the value vector is *fixed*. So given a leaf node, the attention value is independent of the query itself. Note that this requirement completely flips the tree's tendency to map multiple keys to a single leaf node. That is, to optimize the loss function, TF-ATTENTION would prefer all the keys to map to one leaf node, TC-ATTENTION's incentives are completely reversed. That is, if TC-ATTENTION maps all the keys to the same leaf node then it would give an unweighted average of the value vectors of *all* the keys, irrespective of query which is average attention - naturally a weaker model and would have large loss. Instead, TC-ATTENTION would prefer to spread out keys as much as possible in leaf nodes so that it can compute more powerful query dependent attention. We present key distributions of learned TREEFORMER models in Figure 4 supporting this argument.

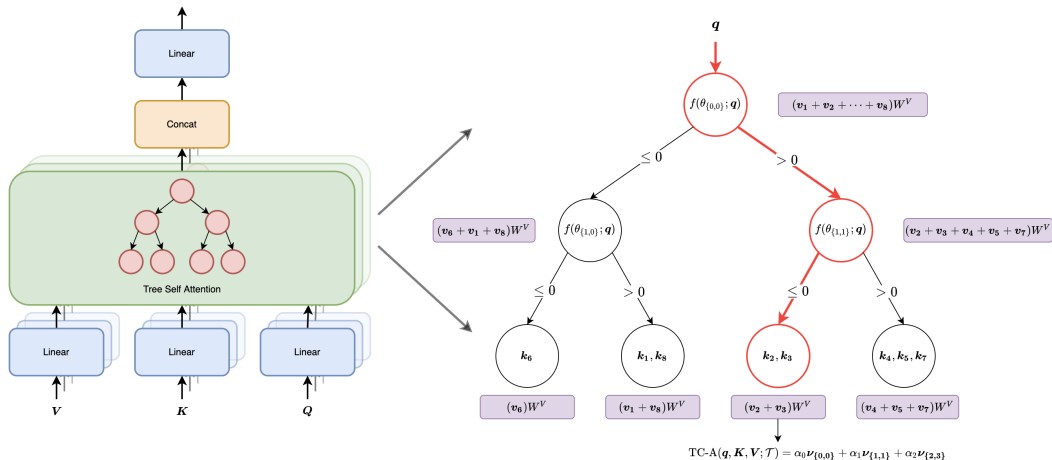

Figure 1: The figure on the left represents the self attention layer in TREEFORMER models, and the tree self attention block is expanded for a representative view of the TC-ATTENTION mechanism.

We further generalize this technique to ensure contributions from other nodes to the value vector. That is, we add contributions to a query's value vector from not only the leaf node but also from the internal nodes that were traversed by the query.

Formally, let $\boldsymbol{\nu}_{l,j}$ be the value vector associated with a node. Then,

$$\boldsymbol{\nu}_{l,j} = \frac{1}{|S_{l,j}(\boldsymbol{K}W^K)|} \sum_{i \in S_{l,j}(\boldsymbol{K}W^K)} \boldsymbol{V}_i W^V,$$

i.e., it is an average of all value vectors of keys such that $l, j$ lies in their paths. Now, given $\boldsymbol{\nu}_{l,j}$ for each node and the tree $\mathcal{T}$ (including it's parameters $\theta$), we get the attention for a query $\boldsymbol{q}$ as:

$$\text{TC-ATTENTION}(\boldsymbol{q}, \boldsymbol{K}, \boldsymbol{V}; \mathcal{T}) = \sum_{l=0}^{h} \alpha_l \cdot \boldsymbol{\nu}_{\{l, P_{\mathcal{T}}(l, (W^Q)^T \boldsymbol{q})\}} \tag{4}$$

where $\alpha_l$ is the learnable attention weight at each level of the decision tree. See Figure 1 for a visual representation of TC-ATTENTION.

Note that as TC-ATTENTION applies query independent averaging, it is easy to design counterexamples where the standard attention layer is strictly more powerful than TC-ATTENTION. Now, we can make TC-ATTENTION as powerful as a standard attention layer by allowing traversal of multiple paths and $\alpha_l$ to be query dependent. However, our results indicate that TC-ATTENTION is sufficiently powerful, and might not require more modeling power in the attention layer. We leave further exploration of multiple traversal paths for future work.

**Computational Complexity** The cost of TC-ATTENTION can be broken down into three parts: 1. projection of query, key, and value tensors - note that we can embed projection matrices for both query and key in the parameters of the tree (with two different sets of tree parameters). So the cost is *half* of standard attention: $cd^2n$ 2. storing projected key vectors in the decision tree - $O((2^{h+1} - 1)d)$, and 3. computing the path and attention layer transformation for all queries $\boldsymbol{Q}$ - $O(ndh)$.

Compared to the full attention mechanism - $O(n^2d + d^2n)$, the computational complexity for TC-ATTENTION is linear in sequence length $n$: $O(ndh + (2^{h+1} - 1)d + d^2n)$. In all our experiments, decision tree height $h$ is generally $\leq 10$. Also, note that compared to TF-ATTENTION, the computational complexity is *guaranteed* to be linear in both $n$ and $h$, irrespective of the trained trees/model.

### 3.3 TRAINING OF TREEFORMER MODELS

**Pre-training** It is well-known that learning tree is a challenging problem due to the discrete structure. We use recent techniques (Karthikeyan et al., 2021) to help with the optimization, and ensure

---

**Algorithm 1:** TREEFORMER 2-D Bootstrapping Algorithm

---

**Input :** $M = M_{\{1,\cdots,L\}}(h_s)$ - TREEFORMER model with height $h_s$, $w$ - layer width for bootstrapping, $h_f$ - height of desired TREEFORMER model, $N$ - number of training steps

**Result:** TREEFORMER model $M_{\{1,\cdots,L\}}(h_f)$

**for** $i = 1$ **to** $\lceil \frac{L}{w} \rceil$ **do**
    idx $= 1 + max(0, L - w \cdot i)$;
    **for** $h = h_s + 1$ **to** $h_f$ **do**
        Initialize $M_{\{\text{idx},\cdots,L\}}(h)$ using parameter weights from $M$. Tree parameters in the attention layer are initialized up to depth $h - 1$ and the remaining leaf node parameters are initialized randomly;
        Train $M_{\{\text{idx},\cdots,L\}}(h)$ for $\frac{N}{\lceil \frac{L}{w} \rceil \cdot (h_f - h_s)}$ steps using self supervised learning;
    **end**
    $M = M_{\{\text{idx},\cdots,L\}}(h_f)$;
**end**
**return**

---

better flow of gradients throughout the network. Despite this, we found in our experiments that pre-training complete TREEFORMER models with tree attentions in all $L$ layers is challenging and often leads to poor sub-optimal solutions. To overcome this challenge issue and avoid inaccurate saddle points, we use a method of incremental bootstrapping for training TREEFORMER (see Algorithm 1). Specifically, we first pre-train an existing model (say a transformer or BigBird). We then introduce tree attention (either TF-ATTENTION or TC-ATTENTION) only in the last few layers (layer $j$ to $L$). The new architecture – denoted by $M_{\{j,\cdots,L\}}$ – is trained for a fixed number of steps using the standard self-supervised learning objective. Now, all the parameters of $M_{\{j,\cdots,L\}}$ are used to initialize the $M_{\{i,\cdots,j,\cdots,L\}}$ model. We do this bootstrapping until we have the complete $M_{\{1,\cdots,L\}}$ TREEFORMER model and it is also trained for a given number of steps. In a similar fashion, we can do this bootstrapping process to initialize TREEFORMER models for a given height using a TREEFORMER model with a smaller tree height. Based on TF-ATTENTION or TC-ATTENTION model, multiple bootstrapping techniques for tree levels can be explored: random assignments in next level, assignment to one node only etc. In this work, we mostly use random assignments based approach where the tree parameters in layers $\{i,\cdots,j-1\}$ are randomly initialized.

**Fine-tuning** We don't do any bootstrapping during the downstream fine-tuning tasks and tree attention exists in all layers of the models. But, we tried two variants of keeping the pre-trained decision tree weights fixed and allowing them to be trained during fine-tuning, and the corresponding results are reported in Section 4. Overall, bootstrapping in finetuning phase leads to similar results as standard fine-tuning with all the model and tree parameters.

## 4 EXPERIMENTS

In this section, we present experimental results comparing the TREEFORMER models with Transformers and with popular efficient Transformer architectures – BigBird (Zaheer et al., 2020), Linformer (Wang et al., 2020) and Performer (Choromanski et al., 2021). We first evaluate our TREEFORMER on the standard NLP benchmark of BERT-based pre-training and *GLUE+SQuAD finetuning* (Devlin et al., 2019). Next, to study effectiveness of TREEFORMER on long sequence length tasks, we present results on the standard benchmark for methods in long-sequence regime: *Long Range Arena* (LRA) (Tay et al., 2021). We also provide *ablation* studies to understand challenges in optimizing TREEFORMER and the utility of the bootstrapping approach (Algorithm 1).

The complete experimental setup along with the hyper-parameters for each of the tasks and hardware details is provided in Appendix A. Please note that we will be using a shorter notation of TF-A for TF-ATTENTION and TC-A for TC-ATTENTION.

### 4.1 BERT PRE-TRAINING / FINE-TUNING

Following Devlin et al. (2019) we use a **masked LM (MLM)** objective to pretrain our TREEFORMER BERT-Base models on Wikipedia, Books (Zhu et al., 2015), CC-News (Guu et al., 2020), and Stories (Trinh & Le, 2018) datasets.

Table 2: BERT Pre-training and Fine-tuning on sequence length 512. We report the accuracies for MLM and MNLI, F1 Score for SQuAD, and the average over tasks for the GLUE Benchmark. We see that TF-A matches the baseline with **3x** reduction in computation, and $k$-ary TF-A improves GLUE score by $> 1\%$! TC-A models with 6 fine-tuned decision tree layers matches the baseline, and models with higher number of tree layers underperform. Please note that Linformer suffers a **1.3%** drop on GLUE and BigBird does *not* offer compute savings for smaller sequence lengths ($\leq 512$).

| Model | MLM | MNLI | SQuAD | GLUE (Avg.) | FLOPs |
|---|---|---|---|---|---|
| BERT-Base | 71.91 | 81.33 | 88.74 | 80.11 | 100% |
| Linformer ($k = 128$) | 71.22 | 81.02 | 85.19 | 78.75 | 49.59% |
| BigBird | 72.28 | 81.67 | 86.24 | 80.16 | 100.40% |
| TREEFORMER TF-A (12 layers) | **73.24** | 82.73 (**83.32**) | 88.69 (87.65) | 80.05 | 29.87% |
| TREEFORMER TC-A (6 layers) | 72.30 | 81.73 (80.98) | 86.55 (85.61) | 80.55 | 57.98% |
| TREEFORMER TC-A (9 layers) | 70.88 | 80.66 (80.40) | 84.32 (83.66) | 79.51 | 36.97% |
| TREEFORMER TC-A (12 layers) | 64.91 | 74.29 (70.28) | 71.96 (54.19) | 71.39 | 15.96% |
| TREEFORMER $k$-ary TF-A | 73.04 | 83.08 | **89.00** | **81.25** | 25.44% |

**Fine-tuning - GLUE/SQuAD** We fine-tune the pre-trained BERT-Base models on 7 datasets from the GLUE benchmark (Wang et al., 2018) including MNLI and the SQuAD (Rajpurkar et al., 2016) dataset for question answering. Table 2 shows the comparison between various architectures on these tasks. We observe that the TREEFORMER model with TF-ATTENTION and $k$-ary TF-ATTENTION work well - beating the BERT baseline in MNLI and SQuAD while using fewer FLOPs. The bracketed values in Table 2 for the TREEFORMER models represent fine-tuning while the decision tree parameters are fixed and obtained from the corresponding pre-trained TREEFORMER models.

We would like to make multiple observations here: a). TF-ATTENTION is competitive with both baseline BERT model as well as the BigBird model. In fact, for MNLI task, it is **1%** more accurate than BERT, and on SQuAD, it is **2%** more accurate than BigBird, despite requiring **3x** fewer FLOPs in the attention layer. b). TF-ATTENTION is more accurate than Linformer as well, while still being cheaper c). TC-ATTENTION provides accuracy values competitive with baseline Linformer, BigBird only when TC-ATTENTION layer is applied in 6 of the 12 transformer blocks. With a larger number of TC-ATTENTION layers, the accuracy drops significantly, confirming challenges in optimizing a difficult discrete layer across multiple layers. We leave further investigation into better optimization of TC-ATTENTION in deep architectures for future work.

**Benefits of Bootstrapping** Figure 2 shows the training progression of the TREEFORMER models with bootstrapping (Algorithm 1). The sudden decrease (increase) in MLM accuracy (loss) happens at the bootstrapping stage, when a model with more decision tree layers is initialized with a model with less decision tree layers. We observe that while bootstrapping causes a momentary drop, it eventually leads to a higher final accuracy for TF-A. For TC-A we notice improvement in bootstrapping upto 6 decision tree layers (till $\approx 1e6$ steps), but going to 12 decision tree layers causes training instabilities.

We next compare the training progress for TREEFORMER models with different decision *tree heights* in Figure 3. We notice that TF-A performs better with bootstrapping for smaller tree heights, but suffers as tree height goes to 7. On the other hand we find TC-A to be much more challenging in bootstrapping to more that 6(out of 12) layers. For larger heights the training is hard due to vanishing gradients and MLM accuracy approaches 0 at the last stage of the bootstrapping process.

**Leaf distribution** Figure 4 shows the distribution of key vectors in the leaf nodes of the TREEFORMER models in specific layers. Lighter lines represent the initial stages of the pre-training process and darker lines represent the distribution at later stages in the pre-training – showing the evolution of distribution as the training progresses. We observe that the distribution is more skewed in later layers compared to the initial layers. Moreover, the distribution is much more skewed in the TF-A variant compared to TC-A. This is because TF-A, being a sparse attention mechanism, will try to cluster more keys in a given leaf node to attend to all relevant keys for a given query (see § 3).

## 4.2 LONG RANGE ARENA (LRA)

We next perform experiments on the Long Range Arena Benchmark (LRA) (Tay et al., 2021). It comprises of 6 tasks (*ListOps*, *Text*, *Retrieval*, *Image*, *Pathfinder*, and *Path-X*) with inputs having long

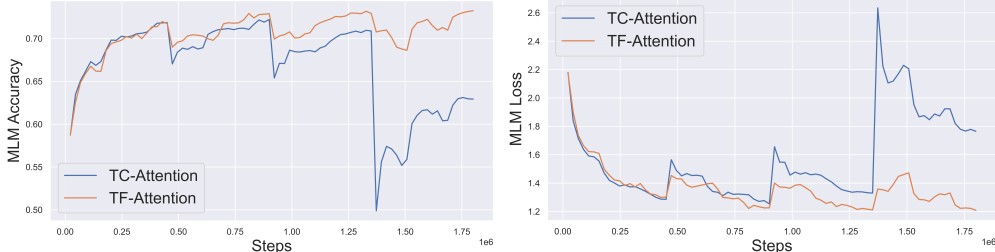

Figure 2: Progression of MLM training for TREEFORMER models with the two tree attention variants. The jump in loss/accuracy corresponds to points in training when bootstrapping was done. This allows us to train TREEFORMER avoiding the optimization issues in training from scratch.

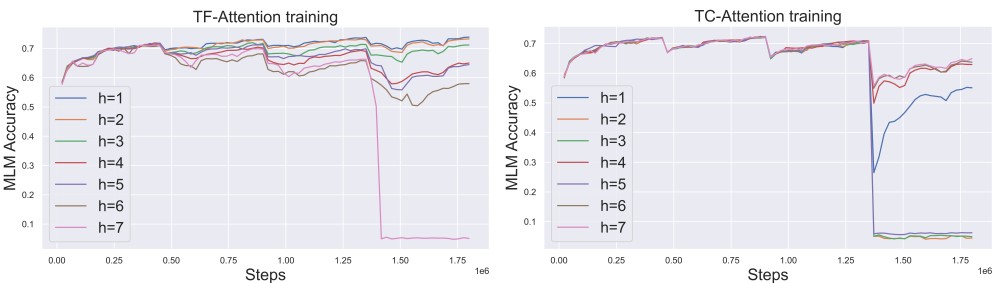

Figure 3: Progression of MLM accuracy while training TREEFORMER models with different decision tree heights. Notice that models with larger tree heights are much more challenging to train and they completely collapse as the bootstrapping process gradually restricts attention.

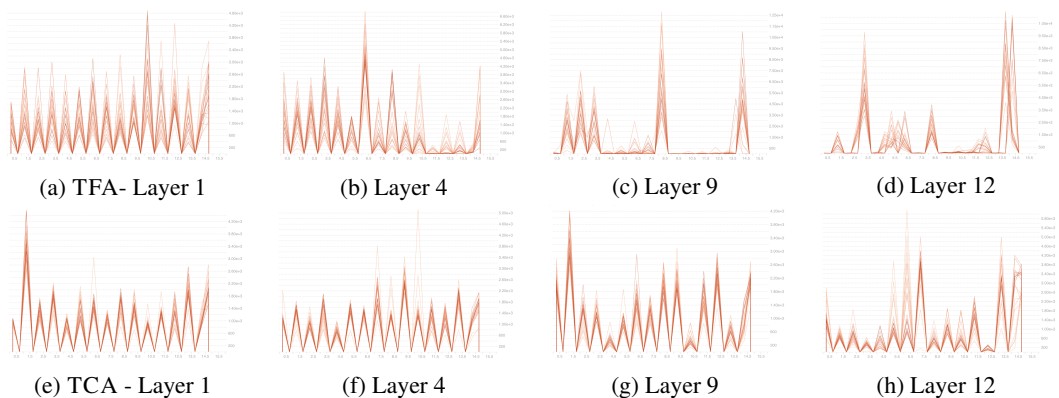

Figure 4: Distribution of key vectors across the leaf nodes of the decision tree in different layers of TREEFORMER- TF-A (a-d) and TC-A (e-h) models. We notice that TC-A helps in training a more uniform key distribution in comparison to TF-A as discussed earlier (see § 3).

sequence lengths ranging from *1024 to 16384*. Three of the tasks are text-based and the remaining three comprise of visual inputs. In our comparison, we skip the *Pathfinder* task in the LRA benchmark because of the very high variance in the accuracy for all models, and also skip *Path-X* because all model architectures fail in this task due to the memory or computational bottlenecks at sequence length 16384. The results of various model architectures across the remaining 4 tasks are presented in Table 3. Along with the performance, we also show the computational FLOPs relative to the standard Transformer averaged over the 4 tasks for each of the models in the self attention layer of an encoder block in the model architecture.

On average, both TF-ATTENTION and TC-ATTENTION are almost as accurate as the baseline Transformer as well as the BigBird. Further, attention layer FLOPs of TF-ATTENTION is almost **3x** lower than that of Transformer. These factors are further improved by using the $k$-ary TF-ATTENTION

Table 3: **LRA Benchmark.** Comparison of TREEFORMER with other efficient transformers. We notice that TREEFORMER models perform as well as baseline and even improve over it in some tasks, while using much less FLOPs. In particular, TC-A matches baseline and improves over other efficient transformers with **33x** reduction in compute. TF-A and $k$-ary TF-A while suffer from imbalanced key allocations, still match the baseline performance and lead to **3x** FLOPs reduction.

| Model | ListOps | Text | Retrieval | Image | *Average* | FLOPs |
|---|---|---|---|---|---|---|
| Transformer | 36.37 | 64.27 | 57.58 | 42.35 | 50.14 | 100% |
| Linformer | 35.70 | 53.94 | 52.93 | 37.92 | 45.12 | 2.57% |
| Performer | 18.01 | 65.40 | 54.56 | 42.51 | 45.12 | 25.37% |
| BigBird | 36.05 | 64.02 | 58.98 | 40.34 | 49.85 | 27.13% |
| TREEFORMER TF-A | 36.02 | 64.38 | 58.52 | 42.31 | **50.31** | 34.18% |
| TREEFORMER TC-A | 35.54 | 64.41 | 58.38 | 42.01 | 50.09 | 3.05% |
| TREEFORMER $k$-ary TF-A | 35.76 | 64.62 | 58.29 | 42.34 | 50.25 | 29.82% |

Table 4: Inference times for different input sequence lengths.

| Model ↓ Seq. Length → | 1024 | 2048 | 4096 | 8192 |
|---|---|---|---|---|
| Transformer | 108ms | 391ms | 1.83s | 9.23s |
| TREEFORMER TF-A | 113ms | 218ms (**1.8x**) | 549ms (**3.3x**) | 1.37s (**6.7x**) |
| $k$-ary TREEFORMER TF-A | 108ms | 230ms (**1.7x**) | 584ms (**3.1x**) | 1.55s (**6x**) |

variant. TC-ATTENTION on the other hand is almost **33x** cheaper than Transformer and **9x** cheaper than BigBird despite achieving almost the same accuracy. Only Linformer has similar computational cost as TC-ATTENTION, but TC-ATTENTION is almost **5%** more accurate than Linformer. In terms of total FLOPs for the transformer encoder block, the self attention comprises of 50.84% of the total computational FLOPs in the *ListOps* task, and the rest is used in dropouts and the two feed forward layers. For *Text*, the distribution is 73.3% and 26.7% respectively. So, we can get a total speedup of 2-3x by using TREEFORMER models instead of the standard Transformer.

**Inference times** In Table 4, we finally present a comparison of inference wall time between the TREEFORMER TF-ATTENTION model and the standard attention on a Intel Xeon Platinum P-8136 CPU using one thread and 100GB of RAM. We measure the inference time for the self-attention module using 8 attention heads and hidden dimension 768 for a batch size of 1. We use a decision tree of height 6 for TF-ATTENTION. Our implementation of TREEFORMER TF-ATTENTION model iterates over the leaf nodes and *gathers* the subset of queries and keys that lie in a given leaf node, and only computes attention scores by doing matrix multiplication between these gathered queries and their relevant keys. The final projections for the *gathered* keys are then *scattered* into the original query matrix. For sequence lengths 2048 - 8192, we get a speedup ranging from **1.8x** - **6.7x**.

## 5    CONCLUSION

In this work we proposed TREEFORMER with a novel efficient attention mechanism that leverages the idea of finding top nearest neighbors using decision trees and applying it to attention computation. We proposed two attention variants TF-ATTENTION and TC-ATTENTION with different sparsity levels. We showed that using techniques like dense gradient trees and bootstrapping allows us to train the TREEFORMER models in an end to end manner with simple back propagation. Using extensive experiments, we showed that TREEFORMER models achieve comparable or better accuracy than other efficient attention models, but with significantly lower computational complexity.

While decision trees are popular in other applications (XgBoost (Chen & Guestrin, 2016), C4.5 (Quinlan, 2014), KD tree (Bentley, 1975)), there have been few works integrating and jointly training them with neural networks. Our work takes a first step in that direction and shows that while optimizing with decision trees can be challenging, it is also rewarding. Some interesting future research directions include developing better optimization techniques for discrete models, efficiently combining decision trees with different types of neural network architectures, and using multiple decision trees to reduce discretization.

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

## A    EXPERIMENTAL SETUP

In this section we give a detailed description of our experimental setup including various hyper-parameters.

### A.1    BERT PRE-TRAINING/FINE-TUNING

Table 5 detail the four public datasets we used for pre-training the TREEFORMER and baseline models. For all datasets, we truncate documents with length greater than 512 to multiple sequences of length 512. We follow the recommendations of Devlin et al. (2019) for MLM pre-training, and mask 15% tokens in each input sequence. We report the MLM performance in terms of accuracy of prediction on the masked positions. We list the hyper-parameters used for pre-training in Table 6. We use Adam optimizer with decoupled weight decay (*AdamW*). We also use a learning rate warmup for the first 10k steps, with linear decay of the learning rate afterwards. We train the TREEFORMER models for a total of 1.8M steps - 450000 steps during each of the four bootstrapped processes shown in Figures 2 and 3. The layer-width for each bootstrapping is 3, meaning 3 tree attention layers are added in every subsequent bootstrapping process.

Table 5: Pre-training Datasets.

| Dataset | # Tokens | Document Length |
|---|---|---|
| Books (Zhu et al., 2015) | 327M | 37K |
| CC-News (Guu et al., 2020) | 3.7B | 561 |
| Stories (Trinh & Le, 2018) | 2.5B | 8.2K |
| Wikipedia | 1.5B | 592 |

Table 6: Hyper-parameters for the two TREEFORMER model variants for BERT Pre-training.

| Parameter | TREEFORMER TF-A | TREEFORMER TC-A |
|---|---|---|
| Tree Height, $h$ | 2 | 7 |
| Sequence Length | 512 | 512 |
| Attention Heads | 12 | 12 |
| Hidden Layers | 12 | 12 |
| Hidden Layer Size | 768 | 768 |
| Batch Size | 1024 | 1024 |
| Activation Layer | gelu | gelu |
| Dropout Prob. | 0.1 | 0.1 |
| Attention Dropout Prob. | 0.1 | 0.1 |
| Optimizer | AdamW | AdamW |
| Learning Rate | $10^{-4}$ | $10^{-4}$ |
| Hardware (TPUv3 slice) | $8 \times 16$ | $8 \times 16$ |

For fine-tuning, we use the same pre-trained models and fine-tune on the downstream tasks with most of the model parameters borrowed from the pre-training process. The parameters that are different for various datasets in the GLUE benchmark and SQuAD are detailed in Table 7. For MNLI, we report the matched accuracy, Matthews Correlation Coefficient (MCC) for COLA, F1 Scores for MRPC, and accuracies for the remaining tasks - QQP, QNLI, SST-2, and RTE. On the SQuAD dataset, we report the F1 scores on v1.1 version of the dataset.

Table 7: Hyper-parameters for BERT Fine-tuning for GLUE and SQuAD.

| Parameter | MNLI, QQP, QNLI, SST-2 | COLA, MRPC | RTE | SQuAD |
|---|---|---|---|---|
| Batch Size | 128 | 16 | 16 | 48 |
| Epochs | 3 | 10 | 10 | 2 |
| Learning Rate | $3 \times 10^{-5}$ | $1 \times 10^{-5}$ | $2 \times 10^{-5}$ | $8 \times 10^{-5}$ |
| Hardware (TPUv3 slice) | $4 \times 4$ | $4 \times 2$ | $4 \times 2$ | $2 \times 2$ |

## A.2 LONG RANGE ARENA (LRA)

The task specific hyper-parameters used for the LRA Benchmark (Tay et al., 2021) are presented in Table 8. We use Adam optimizer for all the experiments but the weight decay factor is different for *Image* task as highlighted in Table 8. We start the bootstrapping for TC-ATTENTION variant of the TREEFORMER from tree height $h = 2$ up to all the way to $h = 10$. We notice that $h = 6$ performs the best on most of the LRA tasks. So we report the performance numbers for $h = 6$. We also include the number of training steps during each bootstrapping process in the table.

Table 8: Hyper-parameters for the various LRA tasks for TREEFORMER models.

| Parameter | Listops | Text | Retrieval | Image |
|---|---|---|---|---|
| Max. Sequence Length | 2048 | 4096 | 8192 | 1024 |
| Tree Height for TREEFORMER | 6 | 6 | 6 | 6 |
| # Layers | 4 | 4 | 4 | 1 |
| Attention Heads | 4 | 4 | 4 | 1 |
| Embedding Dim. | 512 | 256 | 128 | 32 |
| MLP Dim. | 1024 | 1024 | 512 | 64 |
| Batch Size | 32 | 32 | 32 | 56 |
| Dropout Prob. | 0.1 | 0.1 | 0.1 | 0.3 |
| Attention Dropout Prob. | 0.1 | 0.1 | 0.1 | 0.2 |
| Learning Rate | 0.05 | 0.05 | 0.05 | 0.0005 |
| # Training Steps | 5000 | 20000 | 5000 | 35000 |
| Weight Decay for Adam | 0.1 | 0.1 | 0.1 | 0.0 |
| Hardware (TPUv3 slice) | $2 \times 2$ | $2 \times 2$ | $2 \times 2$ | $2 \times 2$ |

## B EXPERIMENTAL RESULTS

In this section, we include some extra results in addition to those present in the main paper.

### B.1 BERT PRE-TRAINING/FINE-TUNING

We first present the task-wise performance of the different models on the GLUE benchmark in Table 9. In this section, we also demonstrate the importance of bootstrapping in the training process with the help of some empirical evidence using the same set of experiments as in the paper. We track the gradient norms during the pre-training process for TREEFORMER models with bootstrapping and TREEFORMER models trained from scratch without any bootstrapping to see how they compare during the pre-training stage. Figure 5 shows the corresponding plots for BERT-based TREEFORMER models in layers $1, 4, 8, 12$ of the models. We observe that gradient norms for the bootstrapped TREEFORMER model are higher compared to the non-bootstrapped version of the same model. This shows the challenges in training decision trees directly with neural networks and the advantage of the proposed bootstrapping approach.

### B.2 LONG RANGE ARENA (LRA)

Table 10 extends Table 3 to include more models like Kitaev et al. (2020), Sun et al. (2021), and Gu et al. (2021). Although Gu et al. (2021) is the state of the art in the LRA benchmark, the paper itself mentions that S4 cannot be scaled to sequence modeling using very large models because of its compute and memory requirements. S4 experiments in the paper are on smaller models and it cannot scale to the level of Transformer-based models involving billions of parameters. Sun et al. (2021) performs better than TREEFORMER models on LRA, but it loses performance on GLUE tasks compared to the baseline Transformer models.

In Table 3, we presented the average speedup (FLOPs column) over the four tasks in the LRA Benchmark for each of the models. Table 11 breaks down that average and details the speedup for each representative task in LRA.

Table 9: GLUE Benchmark

| Model | MNLI | QQP | QNLI | SST-2 | MRPC | RTE | CoLA |
|---|---|---|---|---|---|---|---|
| BERT-Base | 81.33 | 89.62 | 87.93 | 90.63 | 89.55 | 67.19 | 54.53 |
| Linformer ($k = 128$) | 81.08 | 88.74 | 87.19 | 88.56 | 87.96 | 65.33 | 52.38 |
| BigBird | 81.67 | 87.84 | 88.42 | 92.77 | 89.20 | 66.32 | 54.93 |
| TREEFORMER TF-A (12 l) | 83.32 | 89.19 | 87.67 | 89.93 | 88.40 | 66.41 | 55.41 |
| TREEFORMER TC-A (6 l) | 81.73 | 89.75 | 88.86 | 89.58 | 89.47 | 68.75 | 55.70 |
| TREEFORMER TC-A (9 l) | 80.66 | 89.22 | 87.41 | 89.00 | 87.80 | 67.58 | 54.89 |
| TREEFORMER TC-A (12 l) | 74.29 | 85.63 | 82.65 | 84.14 | 84.01 | 62.11 | 26.88 |
| TREEFORMER $k$-ary TF-A (12 l) | 83.08 | 91.24 | 88.30 | 92.84 | 89.43 | 68.75 | 55.08 |

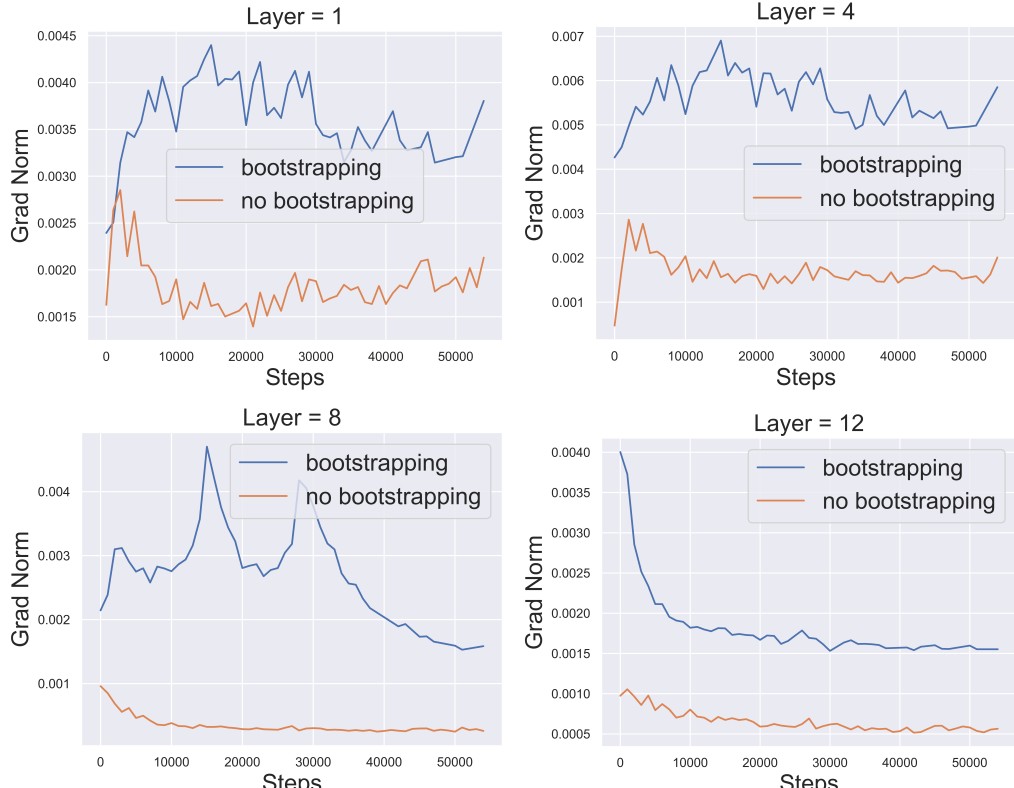

Figure 5: Comparison of gradient norms as the training progresses.

Table 10: **LRA Benchmark** (extended).

| Model | ListOps | Text | Retrieval | Image | *Average* | FLOPs |
|---|---|---|---|---|---|---|
| Transformer | 36.37 | 64.27 | 57.58 | 42.35 | 50.14 | 100% |
| Reformer | 37.27 | 56.10 | 53.40 | 38.07 | 46.21 | - |
| BigBird | 36.05 | 64.02 | 58.98 | 40.34 | 49.85 | 27.13% |
| TREEFORMER TF-A | 36.02 | 64.38 | 58.52 | 42.31 | 50.31 | 34.18% |
| TREEFORMER TC-A | 35.54 | 64.41 | 58.38 | 42.01 | 50.09 | 3.05% |
| TREEFORMER $k$-ary TF-A | 35.76 | 64.62 | 58.29 | 42.34 | 50.25 | 29.82% |
| Luna | 37.98 | 65.78 | 79.56 | 47.86 | 57.80 | - |
| S4 | 59.60 | 86.82 | 90.90 | 88.65 | **81.49** | - |

Table 11: **LRA Benchmark.** Comparison of TREEFORMER with other efficient transformers in terms of FLOPs in the attention layer.

| Model | ListOps | Text | Retrieval | Image | *Average FLOPs* |
|---|---|---|---|---|---|
| Transformer | 100% | 100% | 100% | 100% | 100% |
| Linformer | 2.98% | 2.28% | 0.57% | 4.44% | 2.57% |
| Performer | 28.29% | 14.47% | 6.76% | 51.96% | 25.37% |
| BigBird | 31.98% | 16.35% | 7.74% | 52.43% | 27.13% |
| TREEFORMER TF-A | 37.82% | 28.19% | 29.48% | 41.23% | 34.18% |
| TREEFORMER TC-A | 4.57% | 2.34% | 0.61% | 4.66% | 3.05% |

Table 12: Summarization ROUGE scores obtained for the PubMed and Arxiv datasets. We notice that TREEFORMER models outperform the baseline Transformer and BART models, and perform competitively with BigBird.

| Model | PubMed | | | Arxiv | | | *Average FLOPs* |
|---|---|---|---|---|---|---|---|
| | **R-1** | **R-2** | **R-L** | **R-1** | **R-2** | **R-L** | |
| Transformer | 41.11 | 16.17 | 25.50 | 29.90 | 7.40 | 20.53 | 100% |
| BART | 45.47 | 19.41 | 27.67 | 31.74 | 8.96 | 21.02 | 100% |
| BigBird | 46.69 | 20.36 | 28.27 | 33.72 | 9.43 | 21.55 | 19.25% |
| TREEFORMER TF-A | 45.02 | 19.34 | 26.30 | 33.34 | 9.29 | 21.48 | 28.76% |
| TREEFORMER $k$-ary TF-A | 45.35 | 19.22 | 26.04 | 33.42 | 9.27 | 21.10 | 32.48% |
| TREEFORMER TC-A | 40.23 | 15.02 | 24.78 | 30.14 | 7.86 | 20.95 | 3.09% |

## B.3 SUMMARIZATION

We also perform experiments on sequence-to-sequence (seq2seq) tasks with encoder-decoder models. Following Zaheer et al. (2020), we only introduce our attention mechanism in the encoder and use standard full attention in the decoder, as the number of output tokens is usually small ($\leq 512$) compared to the input. It has been well established that pre-training also helps in seq2seq tasks like summarization, machine translation, and other generative tasks (Lewis et al., 2019; Liu et al., 2020). So, we also pre-train our encoder-decoder TREEFORMER models using the BART objective.

We use the pre-trained encoder-decoder models and finetune on the abstractive summarization task on the PubMed and Arxiv datasets consisting of long documents with a maximum sequence length of 3072 for the input. The results on this task are presented in Table 12. Our TF-ATTENTION models outperform the baseline Transformer and BART models, and are competitive with the BigBird models with significant reduction in FLOPs in the attention layer. The FLOPs in TF-ATTENTION models are higher compared to BigBird because of the skewness in leaf nodes of the decision tree. We leave further optimization of TF-ATTENTION to enable less leaf-node skew for future work.

## C PROOFS

We prove Lemma 1 by construction - we show that there exists a choice of the decision tree weights for all layers of the TREEFORMER TF-ATTENTION model, such that all the queries and keys cluster to the same leaf node.

From Equation 2, we have the tree parameters $(\boldsymbol{w}_{\{l,j\}}; b_{\{l,j\}})$ for tree $\mathcal{T}$. We assign the parameters as follows:

$$
\begin{aligned}
\boldsymbol{w}_{\{l,j\}} &= 0 \quad \forall j = 0, \cdots, 2^l - 1 \quad \forall l = 0, \cdots, h \\
b_{\{l,j\}} &= \epsilon,
\end{aligned}
\tag{5}
$$

where $\epsilon$ is any small positive value. We denote the tree with the above parameters as $\mathcal{T}_0$. This assignment will ensure that all the queries and keys will follow the right-most path in the decision tree, and cluster in the right-most leaf node. Thus, for $\mathcal{T} = \mathcal{T}_0$, we have TF-ATTENTION$(\boldsymbol{Q}, \boldsymbol{K}, \boldsymbol{V}; \mathcal{T}) =$ Attention$(\boldsymbol{Q}, \boldsymbol{K}, \boldsymbol{V})$.

