# OpenReview forum: "Treeformer: Dense Gradient Trees for Efficient Attention Computation"
_ICLR.cc/2023/Conference — ICLR 2023 poster_

### Official Review · Reviewer_xiDb · 2022-10-23

**Confidence:** 4
**Correctness:** 3
**Technical Novelty And Significance:** 3
**Empirical Novelty And Significance:** Not applicable
**Recommendation:** 6

**Clarity, Quality, Novelty And Reproducibility:**

- Clarity: the paper is written clearly.
- Quality: the results on MNLI, GLUE and SQuAD are convincing, but the experiments on LRA is weak.
- Novelty: the proposed method is novel.
- Reproducibility: the code is not provided, but the authors provide hyperparameters for all experiments.

**Strength And Weaknesses:**

## Strengths
- The methods are novel.
- The authors conduct experiments on a lot of benchmark datasets. The results on SQuAD and GLUE are strong.
- The proposed method is novel and it is interesting that the proposed tree-based attention can outperform conventional self-attention

## Weaknesses
- It would be better if the models can include results on generative tasks where casual attention is used such as causal language modeling,
- The author omits stronger baselines on LRA and only includes old inferior models. For example, recent papers including Luna, H-Transformer-1D, S4, and DSS achieve better performance while being efficient.
- Despite saving a significant number of FLOPs, the method cannot be implemented efficiently. As shown in Table 4, Treeformer TF-A starts to be faster than Transformer only after the sequence length gets to 4096. Moreover, I think the authors should include the inference time of Treeformer TF-C, and Treeformer $k$-ary TF-A. Moreover, it would be better to see the inference time comparison between Treeformers and other efficient Transformer variants.

**Summary Of The Paper:**

This paper proposes Treeformer, a modified Transformer that replaces the multi-head self-attention layers with tree-based attention mechanisms. There are three variants studied in this paper: TF-attention (Tree Fine-grained attention), $k$-ary TF-attention, and TC-attention (Tree coarse attention). The authors claim that Treeformer outperforms Transformers while being more efficient (linear time complexity with respect to sequence length). Experimental results on MLM, MNLI, SQuAD, GLUE, and LRA are provided.

**Summary Of The Review:**

In a nutshell, this paper proposes a novel idea and the results on SQuAD and GLUE benchmarks are strong and convincing. Despite its weaknesses in LRA experiments and actual inference time, I believe that the paper is still slightly above the borderline and give it a weak accept.

---

> ### Author Response · Authors · 2022-11-17
> **Response to Reviewer xiDB**
>
> We thank you for your time and effort in reviewing our work and providing thoughtful feedback!
>
> > It would be better if the models can include results on generative tasks where casual attention is used such as causal language modeling
>
> **Response** - We are currently working on including Treeformer attention for decoder-only and encoder-decoder models and corresponding tasks like causal language modeling, summarization/question answering, etc. We do have some preliminary results on long-document summarization involving encoder-decoder models on the PubMed dataset. The ROUGE scores on this task are presented in the following table.
>
> |Model|R-1|R-2|R-L|FLOPs
> |---|---|---|---|---|
> |Transformer|41.11|16.17|25.50|100%|
> |BART|45.47|19.41|27.67|100%|
> |BigBird-RoBERTa|46.69|20.36|28.27|19.27%|
> |Treeformer TF-A|45.02|19.34|26.30|28.76%|
> |$k$-ary Treeformer TF-A|45.35|19.22|26.04|32.48%|
> |Treeformer TC-A|40.23|15.02|24.78|3.09%|
>
> Another point to note is that the Treeformer numbers require hyper-parameter tuning, so the ROUGE scores might increase further. Treeformer TF-A variants outperform the Transformer baseline and match the BART baseline with fewer FLOPs in the attention layer. TC-A variant performs slightly worse than the Transformer baseline but has significantly lesser FLOPs in the attention layer.
>
> > The author omits stronger baselines on LRA and only includes old inferior models. For example, recent papers including Luna, H-Transformer-1D, S4, and DSS achieve better performance while being efficient.
>
> **Response** - We agree with the reviewer that models like S4 are currently the state of the art in LRA, but our main point of comparison is with Transformer-based models because we want to scale the Treeformer attention to the large language model domain. Because of S4’s compute and memory requirements, it cannot be scaled to sequence modeling using very large models. Luna while performing very well on LRA has inferior performance compared to the Transformer baseline on masked language modeling and fine-tuning tasks. We have added the comparison with these models in the appendix section of our paper.
>
> > Despite saving a significant number of FLOPs, the method cannot be implemented efficiently.
>
> **Response** - We have updated our efficient implementation. Our earlier implementation iterated over the queries in a given sequence and computed their projections individually. The numbers reported in the revised version of the paper are computed by iterating over the leaf nodes, and gathering all the queries in a given node, computing their projections simultaneously, and then using the scatter operation to update these projections. This results in a faster implementation of TF-A and k-ary TF-A models with the numbers reported below:
>
> | Model $\downarrow$ Seq. Length $\rightarrow$ | 1024 | 2048 | 4096 | 8192
> |---|---|---|---|---|
> |Transformer|108ms|391ms|1.83s|9.23s|
> |Treeformer TF-A|113ms|218ms (**1.8x**)|549ms (**3.3x**)|1.37s (**6.7x**)|
> |$k$-ary Treeformer TF-A|108ms|230ms (**1.7x**)|584ms (**3.1x**)|1.55s (**6x**)|
>
> We are now able to observe speedups for sequence length >= 2048 and upto 6.7x speedup for sequence length 8192.
>
> We also note that in research, development of new algorithmic approaches should not be limited by existing hardware as in the long run the hardware is evolving and can potentially lead to significant savings in future.

---

> > ### Comment · Reviewer_xiDb · 2022-11-29
> > **Thank you for the clarification**
> >
> > It's great to see that the preliminary encoder-decoder results are promising and that the efficiency of the proposed methods has been improved. Since the major weaknesses of this paper have been either addressed or explained by the authors, I will increase my rating from 6 to 8 (accept).

---

### Official Review · Reviewer_bNoK · 2022-10-25

**Confidence:** 4
**Correctness:** 3
**Technical Novelty And Significance:** 3
**Empirical Novelty And Significance:** 2
**Recommendation:** 6

**Clarity, Quality, Novelty And Reproducibility:**

The paper is written clearly and the idea is novel. But the evaluation (in terms of FLOPs and CPU latency) and baseline comparisons (on LRA) are not consistent.

**Strength And Weaknesses:**


Strength:

1. The idea of using decision trees to cluster the queries and keys is novel to my knowledge. The authors made essential technical contributions (TC-Attention & Bootstrapping) to make their model work.

2. In terms of FLOPs as computational efficiency metrics, the proposed TreeFormers achieve competitive pre-training & fine-tuning performance with significantly reduced FLOPs in attention layers.


Weaknesses:


1. The paper only evaluated their model in short-sequence (512) real NLP tasks and a synthetic long-sequence task. Compared to GLUE, evaluation on real long-sequence NLP tasks such as long-document summarization & long-document question answering [1] is more suitable for this paper.

2. Due to the non-parallelization of the decision-tree design, the paper mainly uses FLOPs and CPU run-time as efficiency metrics. While this is acceptable, when comparing with baselines on LRA, the paper should also include other state-of-the-art non-Transformer models such as S4 [2].

3. The paper misses citation and discussion with other learning-based sparse attention methods [3-5], to which the proposed method belongs.

4. The paper claims the computational complexity for TC-ATTENTION is linear in sequence length n, which is not correct. Notice that the real complexity of TC-Attention is $O(nkdh + (2^{h+1} − 1)d)$, where $k$ is the number of key tokens in the same leaf node as queries. When $h$ is fixed, $k$ increases linearly with $n$, so the complexity is quadratic. When $k$ is fixed, $h$ increases in the linearly with $\log n$, so the complexity is $n \log n$.



[1] Guo, Mandy, Joshua Ainslie, David Uthus, Santiago Ontañón, Jianmo Ni, Yun-Hsuan Sung, and Yinfei Yang. "LongT5: Efficient Text-To-Text Transformer for Long Sequences."

[2] Gu, Albert, Karan Goel, and Christopher Re. "Efficiently Modeling Long Sequences with Structured State Spaces." In International Conference on Learning Representations. 2021.

[3] Kitaev, Nikita, Lukasz Kaiser, and Anselm Levskaya. "Reformer: The Efficient Transformer." In International Conference on Learning Representations. 2019.

[4] Sun, Zhiqing, Yiming Yang, and Shinjae Yoo. "Sparse Attention with Learning to Hash." In International Conference on Learning Representations. 2021.

[5] Wang, Ningning, Guobing Gan, Peng Zhang, Shuai Zhang, Junqiu Wei, Qun Liu, and Xin Jiang. "ClusterFormer: Neural Clustering Attention for Efficient and Effective Transformer." In Proceedings of the 60th Annual Meeting of the Association for Computational Linguistics (Volume 1: Long Papers), pp. 2390-2402. 2022.

**Summary Of The Paper:**

The paper proposes a new sparse attention mechanism that uses decision trees to cluster the queries and keys such that the queries only attend to the keys that fall into the same leaf. To mitigate the collapsing problem (all keys and queries are allocated to one leaf node) in the proposed TF-Attention (Tree Fine-grained Attention), the paper proposes two mitigations: 1) Tree Coarse Attention, where the attention is summed unweighted for the keys in the leaf node. 2) a boot-strapping training strategy that gradually increases the tree height.


**Summary Of The Review:**

The paper proposes a new sparse attention mechanism that uses decision trees to cluster the queries and keys. However, there are several problems in the paper (See Weaknesses). Therefore, I believe the paper is below the acceptance threshold.

---

> ### Author Response · Authors · 2022-11-17
> **Response to Reviewer bNoK**
>
> We really appreciate your time and effort in providing a great review for our paper!
>
> > The paper only evaluated their model in short-sequence (512) real NLP tasks and a synthetic long-sequence task. Compared to GLUE, evaluation on real long-sequence NLP tasks such as long-document summarization & long-document question answering [1] is more suitable for this paper.
>
> **Response** - Please note that we used popular long sequence benchmark LRA, with sequence lengths up to 8k, and pre-training BERT for our initial evaluation following other efficient transformer papers, e.g. BigBird. Thanks for the suggestions, we are indeed currently working on including more tasks like causal language modeling, summarization, and question answering for the Treeformer attention mechanism especially for decoder-only and encoder-decoder based models, and will include the results in the final version. We do have some preliminary results on long-document summarization on the PubMed dataset. The ROUGE scores on this task are presented in the following table.
>
> |Model|R-1|R-2|R-L|FLOPs
> |---|---|---|---|---|
> |Transformer|41.11|16.17|25.50|100%|
> |BART|45.47|19.41|27.67|100%|
> |BigBird-RoBERTa|46.69|20.36|28.27|19.27%|
> |Treeformer TF-A|45.02|19.34|26.30|28.76%|
> |$k$-ary Treeformer TF-A|45.35|19.22|26.04|32.48%|
> |Treeformer TC-A|40.23|15.02|24.78|3.09%|
>
> Another point to note is that the Treeformer numbers require hyper-parameter tuning, so the ROUGE scores might increase further. Treeformer TF-A variants outperform the Transformer baseline and match the BART baseline with fewer FLOPs in the attention layer. TC-A variant performs slightly worse than the Transformer baseline but has significantly lesser FLOPs in the attention layer.
>
> > when comparing with baselines on LRA, the paper should also include other state-of-the-art non-Transformer models such as S4 [2]
>
> **Response** - We agree with the reviewer that S4 is the state of the art for the synthetic LRA task, but in this paper, our main point of comparison is with Transformer-based models because we want to scale the Treeformer attention to the large language model domain. Because of S4’s compute and memory requirements, it cannot be scaled to sequence modeling using very large models (as mentioned in the S4 paper itself). We have added the comparison with S4 and other models like Luna in the appendix section of our paper.
>
> > The paper misses citation and discussion with other learning-based sparse attention methods [3-5], to which the proposed method belongs.
>
> **Response** - We did not add the Reformer model because of its inferior performance on the LRA task. Except for the Listops task, Reformer performs worse than both the Transformer baseline and other efficient attention mechanisms like BigBird. Reformer’s average on the 4 LRA tasks discussed in the paper is 46.21 compared to Transformer’s 50.14, BigBird’s 49.85, and Treeformer TF-A’s 50.31. As for the other models, we have cited and added these in Section 2 of the paper. We have also added Reformer and Luna LRA performance numbers in the appendix section of our paper.
>
> > The paper claims the computational complexity for TC-ATTENTION is linear in sequence length n, which is not correct.
>
> **Response** - The $k$ factor will not come into the picture because once the key vectors are stored in the leaf nodes of the tree taking up $O(ndh)$ time, they are no longer required for individual queries because we are not computing the dot products with the individual key vectors lying in the same leaf node as the query for TC-A. The $O((2^(h+1) - 1)d)$ factor in the cost absorbs the computation of the value vectors in each of the nodes of the tree. So, we just need to traverse the path for a given query and accumulate the weighted value vectors along its path as illustrated in Figure 1 leading to $O(ndh)$ time.

---

> > ### Comment · Reviewer_bNoK · 2022-11-27
> > **Thank you for your response**
> >
> > I thank the authors for additional (some promised) experimental results and clarifications.
> >
> > > We do have some preliminary results on long-document summarization on the PubMed dataset. The ROUGE scores on this task are presented in the following table.
> >
> > Thank you for the additional results. I hope to see the full results (e.g., HotpotQA NaturalQ TriviaQA WikiHop for QA, and Arxiv PubMed BigPatent for summarization as in Big Bird) in the final version.
> >
> > > So, we just need to traverse the path for a given query and accumulate the weighted value vectors along its path as illustrated in Figure 1 leading to O(nhd) time.
> >
> > Thank you for the clarification. I find that TC is actually of linear complexity, though TF is the more performant model in the paper.
> >
> > Overall, I thank the authors for further improving the submission and would like to raise my score to 6.

---

### Official Review · Reviewer_3zNR · 2022-10-26

**Confidence:** 3
**Correctness:** 3
**Technical Novelty And Significance:** 4
**Empirical Novelty And Significance:** 3
**Recommendation:** 8

**Clarity, Quality, Novelty And Reproducibility:**

The idea presented in the paper is novel to address the quadratic complexity of the attention layer respect to the sequence length. Extensive evaluations on both GLUE and LRA and benchmarking against multiple SOTA models show a convincing result for the proposed method TREEFORMER. Overall, it is a high quality paper.

I do have minor concerns on the missing details of boost strapping method to reproduce the experimental results.

**Strength And Weaknesses:**

Strength:
1. This paper proposes a new architecture called TREEFORMER to use decision trees to efficiently compute attention by only retrieving the top nearest neighboring keys for a given query. The TREEFORMER comes with two novel attention mechanisms named TF-ATTENTION and TC-ATTENTION.

2. TREEFORMER is evaluated extensively on both GLUE and LRA (long sequences data) benchmarks against many SOTA models such as Big Bird and Performer to show that TREEFORMER architecture can be almost as accurate as baseline Transformer while using 30x lesser FLOPs in the attention layer. Thus the result shown is convincing.

3. Ablation studies give good insight on why bootstrapping method works well by showing the gradient norms.

Weakness:
My main questions are in the boost strapping method.
1. With boost strapping and learning different heights of trees (specified in Algorithm 1), how much longer of the pre-training process for TREEFORMER require compared with standard transformer models? It mentions that "we first pre-train an existing model (say a transformer or BigBird)". Does it pre-train the existing model for the same number of steps as a regular transformer's pre-training process and then trigger boost strapping process? Knowing this will help readers to better understand if there is any cost / limitation for using TREEFORMER to achieve SOTA FLOPS reduction in the attention layer.

2. Lack of justification on why tree attention is added backwardly starting from the last layer? what happen if you add the tree attention layer forwardly starting from the beginning layer? It would be interesting to see such comparison in the ablation study.

3. What is the configuration of hyper-parameter layer width for boost strapping? (couldn't find it in the appendix as well) and the previously mentioned training steps of pre-training an existing model before boost strapping. Those missing information is required for researchers to reproduce the experimental results.

Some minor comments/questions:
1. It would be better to explicitly mention that there is no pre-training in the LRA. So that readers can better understand the setting of training TREEFORMER.

2. In the computational complexity -- point 2, in section 3.2, should there be n multiplied in front of the complexity value as there are n key vectors to store where n is sequence length?

**Summary Of The Paper:**

Attention computation of Transformers scales quadratically (n^2) with the input sequence length (n), making it a key bottleneck
in scaling Transformers to long inputs. Targeting at such limitation, this paper proposes a new architecture called TREEFORMER to use decision trees to efficiently compute attention by only retrieving the top nearest neighboring keys for a given query. In addition, to make the decision trees to be trained as part of neural networks using back propagation, this paper proposes a novel bootstrapping method to gradually sparsify attention computation. Finally, this paper extensively evaluates TREEFORMER on both GLUE and LRA benchmarks to show that TREEFORMER architecture can be almost as accurate as baseline Transformer while using 30x lesser FLOPs in the attention
layer.

**Summary Of The Review:**

Attention computation of Transformers scales quadratically (n^2) with the input sequence length (n), making it a key bottleneck
in scaling Transformers to long inputs. Targeting at such limitation, this paper proposes a new architecture called TREEFORMER to use decision trees to efficiently compute attention by only retrieving the top nearest neighboring keys for a given query. In addition, to make the decision trees to be trained as part of neural networks using back propagation, this paper proposes a novel bootstrapping method to gradually sparsify attention computation. Finally, the paper conducts an extensive evaluations on both GLUE and LRA and benchmarking against multiple SOTA models, show a convincing result for the proposed method TREEFORMER. Overall, it is a high quality paper.

---

> ### Author Response · Authors · 2022-11-17
> **Response to Reviewer 3zNR**
>
> We appreciate your time and effort in reviewing our work and providing a great feedback!
>
> > how much of the pre-training process for TREEFORMER require compared with standard transformer models? It mentions that "we first pre-train an existing model (say a transformer or BigBird)". Does it pre-train the existing model for the same number of steps as a regular transformer's pre-training process and then trigger boost strapping process?
>
> **Response** - The BigBird/Transformer pre-trained models are not used for bootstrapping Treeformer models. Treeformer is bootstrapped from a random initialization. As for the training time - Treeformer models are trained for the same number of steps. Transformer and family are trained for 1.8M steps and Treeformer models are trained for 4 bootstrapping processes each comprising 450k steps. We do note that accuracy gains for the Transformer models are very incremental going from 450k to 1.8M steps, so they could have been trained for a lesser number of steps.
>
> > Lack of justification on why tree attention is added backwardly starting from the last layer? what happen if you add the tree attention layer forwardly starting from the beginning layer?
>
> **Response** - We experimented with both variants, and found that adding tree attention from the back either matches or performs slightly better than adding tree attention from the front. This might be because of the skewness of distribution in later layers as observed in TF-A (Figure 4),  which suggests that it is better to train more difficult distributions first and for longer. We will add experiments comparing the two in the revised version of the paper.
>
> > What is the configuration of hyper-parameter layer width for boost strapping?
>
> **Response** - Layer width for bootstrapping is 3, and as addressed in the first point, the bootstrapping is done from scratch i.e. from a random initialization. This layer width results in 4 bootstrapping phases (layers 1 - 12) which is also evident in Figures 2 and 3. Thanks for pointing it out, we have explicitly added it in the updated rebuttal version of the paper.
>
> > In the computational complexity -- point 2, in section 3.2, should there be n multiplied in front of the complexity value as there are n key vectors to store where n is sequence length?
>
> **Response** - Once the key vectors are stored in the leaf nodes of the tree taking up $O(ndh)$ time, they are no longer required for individual queries because we are not computing the dot products with the individual key vectors lying in the same leaf node as the query for TC-A. So, for a given query, we just need to know the path followed by the query in the tree and accumulate the node values at each of the nodes as displayed in Figure 1. The accumulation and computation at each node of the tree before passing the queries takes $O((2^{h + 1} - 1)d)$ time.

---

### Author Response · Authors · 2022-11-17
**Common response to all reviewers**

We thank all the reviewers for their detailed and thoughtful feedback. We have incorporated suggestions by the reviewers in our updated manuscript, and we also wanted to emphasize a change in the experiments section of the results, specifically the inference times in Table 4. Our earlier implementation iterated over the queries in a given sequence and computed their projections individually. The numbers reported in the revised version are computed by iterating over the leaf nodes, and gathering all the queries in a given node, computing their projections simultaneously, and then using the scatter operation to update these projections. This results in a faster implementation of TF-A and k-ary TF-A models with the numbers reported below:

| Model $\downarrow$ Seq. Length $\rightarrow$ | 1024 | 2048 | 4096 | 8192
|---|---|---|---|---|
|Transformer|108ms|391ms|1.83s|9.23s|
|Treeformer TF-A|113ms|218ms (**1.8x**)|549ms (**3.3x**)|1.37s (**6.7x**)|
|$k$-ary Treeformer TF-A|108ms|230ms (**1.7x**)|584ms (**3.1x**)|1.55s (**6x**)|

---

### Decision · Program_Chairs · 2023-01-20

**Decision:**

Accept: poster

**Justification For Why Not Higher Score:**

good paper

**Justification For Why Not Lower Score:**

na

**Metareview: Summary, Strengths And Weaknesses:**

The paper proposes TREEFORMER to learn and use decision trees to efficiently compute attention by only retrieving the top nearest neighboring keys for a given query. It uses a bootstrapping method to gradually sparsify attention computation. Reviewers generally likes the idea and the experiments. The authors also clarified some of their (and my) concerns during the rebuttal. I am in favour of accepting this paper.

**Note From Pc:**

if the above contains the word "oral" or "spotlight" please see: "oral" presentation means -> notable-top-5% and "spotlight" means -> notable-top-25%. As stated in our emails, we are disassociating presentation type from AC recommendations

**Summary Of Ac-Reviewer Meeting:**

NA